# Adaptive Divergence under Gene Flow along an Environmental Gradient in Two Coexisting Stickleback Species

**DOI:** 10.3390/genes12030435

**Published:** 2021-03-18

**Authors:** Thijs M. P. Bal, Alejandro Llanos-Garrido, Anurag Chaturvedi, Io Verdonck, Bart Hellemans, Joost A. M. Raeymaekers

**Affiliations:** 1Faculty of Biosciences and Aquaculture, Nord University, N-8049 Bodø, Norway; joost.raeymaekers@nord.no; 2Department of Organismic & Evolutionary Biology, Harvard University, Cambridge, MA 02138, USA; allanosgarrido@fas.harvard.edu; 3Department of Ecology and Evolution, University of Lausanne, Biophore Building, 1015 Lausanne, Switzerland; anurag.chaturvedi@unil.ch; 4Laboratory of Biodiversity and Evolutionary Genomics, KU Leuven, B-3000 Leuven, Belgium; Iomessanga@gmail.com (I.V.); bart.hellemans@kuleuven.be (B.H.)

**Keywords:** landscape genomics, local adaptation, population genetics, species-specific properties, three-spined stickleback, nine-spined stickleback

## Abstract

There is a general and solid theoretical framework to explain how the interplay between natural selection and gene flow affects local adaptation. Yet, to what extent coexisting closely related species evolve collectively or show distinctive evolutionary responses remains a fundamental question. To address this, we studied the population genetic structure and morphological differentiation of sympatric three-spined and nine-spined stickleback. We conducted genotyping-by-sequencing and morphological trait characterisation using 24 individuals of each species from four lowland brackish water (LBW), four lowland freshwater (LFW) and three upland freshwater (UFW) sites in Belgium and the Netherlands. This combination of sites allowed us to contrast populations from isolated but environmentally similar locations (LFW vs. UFW), isolated but environmentally heterogeneous locations (LBW vs. UFW), and well-connected but environmentally heterogenous locations (LBW vs. LFW). Overall, both species showed comparable levels of genetic diversity and neutral genetic differentiation. However, for all three spatial scales, signatures of morphological and genomic adaptive divergence were substantially stronger among populations of the three-spined stickleback than among populations of the nine-spined stickleback. Furthermore, most outlier SNPs in the two species were associated with local freshwater sites. The few outlier SNPs that were associated with the split between brackish water and freshwater populations were located on one linkage group in three-spined stickleback and two linkage groups in nine-spined stickleback. We conclude that while both species show congruent evolutionary and genomic patterns of divergent selection, both species differ in the magnitude of their response to selection regardless of the geographical and environmental context.

## 1. Introduction

The role of gene flow on the evolution of wild populations is a topic that has received substantial attention from biologists for over five decades [1,2]. Evidence from theoretical models as well as empirical studies show that gene flow has the potential to either enhance or disrupt local adaptation, specifically through the distribution of advantageous alleles or by homogenisation of the gene pool [3]. In addition to being essential for our comprehension of evolution in general, understanding the role of gene flow in local adaptation has become exceedingly important for biodiversity conservation in a time of increasing anthropogenic influence on the natural world [4,5]. Targeted gene flow for instance, the method of translocating individuals with predicted advantageous alleles to populations with low genetic diversity, has been getting more attention as a strategy in conservation efforts [6,7]. This conservation approach may become essential in situations of strong habitat fragmentation in which unassisted gene flow has become impossible. However, it remains challenging to reliably predict how wild populations respond to various levels of gene flow as this can be dependent on the initial levels of genetic diversity within the metapopulation, historical distribution patterns and species-specific properties [8]. It is therefore important to study adaptive divergence in systems in which population connectivity and environmental differences have been characterised, as this allows us to make more reliable predictions about the role of gene flow in local adaptation.

Studies on the relative contribution of gene flow on the emergence of adaptive divergence show disparate results and reveal that species’ evolutionary responses can vary considerably. Populations may show evident signatures of local adaptation regardless of gene flow [9,10,11,12]. The maintenance of adaptive variation under gene flow is possible in situations where the selection for locally adapted alleles is stronger than the influx of non-locally adapted alleles. Here, the specific genomic architecture of a species likely also plays a substantial role, for instance, in the case of large effect loci that are clustered together [3]. However, the homogenizing effects of gene flow are often still observed at relatively small spatial scales, allowing for high migration rates even across strong environmental gradients [13,14,15,16]. In such cases, it is expected that the strength of adaptive divergence increases with distance.

Empirical studies and theoretical models addressing this question often focus on a single species along a single spatial scale or environmental contrast [9,10,17,18,19]. This approach can provide novel insights for a species in a specific geographical context, but also has its limitations. First, as the strength of adaptive divergence is affected by both divergent selection and the level of spatial isolation, it is challenging to discern the relative contribution of these two factors. A more direct approach would be to have a study design where environmental variation and distance among sites are not intrinsically confounded. Second, the distribution of genomic and phenotypic variation is affected by the current properties of the landscape, but is also a result of historical events and species-specific properties [20,21]. Here, the study of phenotypic and genetic variation among populations of multiple coexisting species helps us to infer both shared and unique features of population divergence, enabling broader conclusions regarding the dynamics between gene flow and selection across the landscape [22].

In this study, we investigate the spatial and environmental drivers of population divergence in two ecologically similar stickleback species along both a small-scale and large-scale brackish water–freshwater transition. The three-spined stickleback (*Gasterosteus aculeatus*) and the nine-spined stickleback (*Pungitius pungitius*) are phylogenetically related fishes and thus are excellent model species for a comparative analysis of population structure [22,23]. Both species are euryhaline and share a short similar life cycle with often only one breeding season, and their ecological [24,25], behavioural [26] and genomic [27,28,29,30] properties have been studied extensively. Phylogeographic studies show that the species have different ancestral environments. Three-spined stickleback ancestry can be traced back to mainly marine and coastal areas [31], while the nine-spined stickleback has mainly evolved in freshwater [32].

Both species of stickleback are sympatric in the coastal lowlands and upland rivers of Belgium and the Netherlands (Figure 1A). This allows us to study adaptive divergence under gene flow in each species under comparable conditions. Specifically, we selected sampling sites that vary in environmental conditions as well as in spatial connectivity, including both nearby lowland brackish water (LBW) and lowland freshwater sites (LFW), as well as more isolated upland freshwater (UFW) sites.

Importantly, the contrasts between these three sites form the sides of a triangle (Figure 1B) for which we anticipate varying degrees of the relative effects of selection and gene flow. We expect selection to be the predominant evolutionary force between the populations from the LBW and LFW sites, but also between the populations from the LBW and UFW sites. Yet, the response to selection might depend on levels of gene flow, which can either constrain or fuel adaptive divergence, and which we anticipate to be lower between LBW and UFW sites than between LBW and LFW sites. Furthermore, given their similar selective environments, we expect selection to be weak between the populations from the LFW and UFW sites. Finally, in order to locate selective processes within the evolutionary history of each species, we also investigate whether patterns of selection can be mostly attributed to the older or younger branches of the population genealogy of the two species.

## 2. Materials and Methods

### 2.1. Study Area and Sampling Design

Belgium and the Netherlands harbour diverse brackish and freshwater habitats, including estuaries, creeks, rivers, ditches and ponds. The connectivity with the open sea and between these sites is variable. This study includes data from eleven sampling locations, of which eight have been the focus of previous research [22,33]. These eight sites are all located in the Belgian-Dutch coastal lowlands and comprise four LBW (L01, L02, L05, L06) and four LFW sites (L10, L11, L12 and U01). These sites were sampled in the spring of 2009. The new sites for this study include three UFW sites (U10, U11 and U12) and are all located in the upland area of Belgium. These sites were sampled in the spring of 2012. Full sampling procedures are described in detail in [22,33]. In short, a minimum of 24 individuals per species were obtained using a dipnet. Sticklebacks were killed with a lethal dose of MS222 following directions of the KU Leuven Animal Ethics Committee (https://admin.kuleuven.be/raden/en/animal-ethics-committee), and flash-frozen in dry ice. The salinity of the water was determined using a Hach field-monitoring kit at different dates throughout the year. Sites with consistent conductivities < 1000 µS/cm were classified as freshwater sites. For each site, we determined the shortest Euclidean distance to the coast (DTC). 

### 2.2. Morphological Characterisation

We scored fifteen morphological traits in both species including standard length, four armour traits, five body traits, and five gill traits. These fifteen morphological traits were recorded by carrying out linear measurements and trait counts. In addition, we performed geometric morphometric analysis to quantify differences in body shape.

The fish bodies were thawed on ice, measured for body size (standard length (SL); ± 0.1 cm), and weighed (0.01 g). Subsequently, the left side of each individual was photographed next to a linear scale using a standard camera position. After photographing, the caudal fin was collected and stored in 100% ethanol. All bodies were then stored on a 4% formalin solution. After 2 months, the formalin processed bodies were rinsed with water for 72 h and then bleached for 4 h using a 1% KOH bleach solution. After bleaching, the fish bodies were stained using an Alizarin Red solution to facilitate plate count and the characterisation of gill raker morphology. After staining, the number of lateral plates (Plates) on the left side was determined. The presence of a keel, a small modification of the caudal lateral plates, was noted, but not included in the plate count. Subsequently, the length of the pelvic plate (PP), the left pelvic spine (PS) and the first dorsal spine (DS) were measured using a digital caliper (±0.01 mm). Body depth (BD), the diameter of the eye (EYE), dorsal fin length (DF), anal fin length (AF) and tail length (Tail) were measured digitally using the TPS software v.2.18 [34]. Finally, the gill cover was removed to dissect the left part of the gills. With the aid of a stereomicroscope, the number of large gill rakers (NLGR) on the frontal and distal part of the first gill arch was determined. The length of the first branchial arch (GA), as well as the length of its second (LGR2), third (LGR3) and fourth (LGR4) gill raker, were measured under a stereomicroscope.

Variation in body shape was characterised based on geometric morphometric analysis following Sharpe et al., 2008 [35]. For both species, a total of fifteen homologous landmarks (including 12 landmarks and 3 semi-landmarks; Appendix A) were placed on the photograph using the TPS software v.2.18 [34]. We used (semi-)landmarks 1, 7 and 15 to perform digital unbending of the landmark coordinates to correct for potential bending of the caudal fin when the photographs were taken. The fifteen (semi-)landmarks were then transformed using a least-square Procrustes superimposition, resulting in 26 relative warp (RW1–RW26) scores per individual.

### 2.3. DNA Extraction, Genotyping-by-Sequencing and SNP Filtering

A total of 264 individuals per species (i.e., 24 individuals per site and species) were selected for sequencing. Fin clips were used for genomic DNA extraction using the Nucleospin 96 Tissue DNA Extraction kit (Macherey-Nagel, Düren, Germany) according to the manufacturer’s protocol. The DNA extracts were then treated with the methylation-sensitive restriction enzyme *ApeKI* (GCWGC) and subsequently a unique and common barcode adapter was attached by ligation. All samples were pooled, purified and size-selected using a PCR reaction with Illlumina (San Diego, CA, USA) primers. Single-nucleotide polymorphisms (SNPs) were generated using genotyping-by-sequencing [36] on an Illumina HiSeq 2000 sequencing platform generating paired-end 100 base pair reads. SNP genotyping was performed using the TASSEL-GBS v5.2.52 [37] pipeline by setting the restriction enzyme-e *ApeKI*, requiring a minimum tag output of -c 5 and k-mer length between 20 and 64. In running the pipeline, converted sequencing read tags of the three-spined and nine-spined stickleback that were saved in the SQLite database were aligned to their respective reference genomes [29,38] using Bowtie v2.3.5.1 [39]. Alignment rates for the three-spined and nine-spined stickleback were 85.54% and 95.09%, respectively. Individuals with less than 500,000 reads were removed from the database. SNPs from the database were subsequently converted to a Variant Call Format (VCF). 

Further SNP filtering was performed using VCFtools v0.1.13 [40] and we removed SNPs based on read depth (RD < 5), genotype quality (GQ < 20), non bi-allelic, variant coverage (≥0.9), minimum allele frequency (<0.05), linkage disequilibrium (r2 ≥ 0.8; -geno-chisq) and heterozygosity (H_O_ > 0.5; removing potential paralogs). These filtering criteria were equal to the SNP filtering as applied in Raeymaekers et al. 2017 and were chosen to decrease the false positive rate while maintaining a number of SNPs that allowed for population genomics analyses at a high resolution of genomic data. Finally, we retained 10,836 SNPs in 239 individuals and 15,033 SNPs in 241 individuals for three-spined and nine-spined stickleback, respectively (Table 1). We used PGDSpider v2.1.1.5 [41] together with population definition files for the conversion to other data formats, as well as to assign population labels to each individual.

### 2.4. Population Structure and Genetic Diversity Statistics

We assessed population structure using the Bayesian approach implemented in fastStructure v1.0 [42]. For both species, we ran the algorithm for K = 1 to K = 11 under a simple prior using a default starting seed of 100. The generated output was subsequently used by StructureSelector v1.0 [43] to find the most likely population structure based on the maximal marginal likelihood, and to generate structure barplots.

Further downstream analyses were conducted in R 3.6.3 [44], specifically making use of the options and functions under the Hierfstat v0.4.22 [45] and Adegenet v2.1.2 [46] packages. First, we assessed the genetic diversity per species and site by calculating the expected heterozygosity (H_E_). Simple linear regression was used to test for the significance of the decrease of H_E_ with distance to the coast. Overall and pairwise *F*_ST_ values (N = 55 pairwise combinations) were calculated using Adegenet v2.1.2 [46]. In order to test for isolation by distance, the correlation between pairwise *F*_ST_ and geographical distances along waterways among the sites was tested using a simple Mantel test [47] with 9999 permutations. Finally, we performed two-dimensional classical multidimensional scaling (MDS) on the pairwise *F*_ST_ values as an additional way of visualizing population structure.

### 2.5. Signatures of Adaptive Divergence

In order to compare the level of phenotypic differentiation directly with the level of genetic differentiation in each species, we calculated *P*_ST_, an index that quantifies the proportion of population phenotypic variance in quantitative traits [48,49]. *P*_ST_ values along with 95% Bayesian confidence intervals were estimated following [48]. Specifically, traits were assumed to be normally distributed, and a linear model was fitted to each trait separately. Population was included in the model as a random effect, and body size as a covariate. The models were fitted to the data using a Gibbs sampler, implemented in WinBUGS v1.4.3 [50]. Prior distributions for each trait were uninformative, and posterior distributions were obtained by running five independent chains (50.000 iterations) after a burn-in of 1000 iterations.

We utilised two distinct methods for identifying genomic signatures of selection. First, we used BayeScan v2.1 [51] with the prior odds of neutrality set at 100 and starting from an initial 10 pilot runs of 5000 iterations followed by an additional 150,000 iterations and a burn-in of 50,000 iterations. For both species, these settings were applied globally, using all 11 sites, as well as for the three possible combinations of the three spatial scales, i.e., LBW–LFW (8 sites), LBW–UFW (7 sites), and LFW–UFW (7 sites). In each analysis, BayeScan v2.1 considered the included populations separately. The posterior probabilities calculated by BayeScan 2.1 were transformed into q-values corresponding to the FDR (False Discovery Rate) of the *P*-value, and the cut-off for determining statistical significance was set to 0.05.

Second, we used GRoSS v1.0 [52] to assign signatures of positive selection to the branches of the population genealogy, which for each species of stickleback corresponded to the three spatial groups (LBW, LFW, UFW). Specifically, GRoSS v1.0 aims to attribute genomic signatures of positive selection to the branches of an admixture graph. To do so, GRoSS v1.0 uses population phylogenies to determine allele frequencies across hierarchical groups, and tests which of those deviate from what is expected given the population genealogy. In addition, GRoSS v1.0 indicates along which specific evolutionary lineages these allelic variants were most likely selected. Using this method, we were able to identify the relative importance of local and regional selection pressures in the two species. The population phylogeny of each species was inferred using a neighbour-joining tree based on the first two dimensions of a multidimensional scaling analysis. Individuals within populations were bootstrapped 1000 times to generate a consensus tree based on the clusters with the most support. For both species, the consensus tree corresponded to the three spatial groups (LBW, LFW, UFW), and was in agreement with the Bayesian analysis of population structure using fastStructure v1.0 [42].

## 3. Results

### 3.1. Morphological Divergence

Across the full geographical scale of our study (i.e., all eleven locations), populations of the three-spined stickleback were morphologically more divergent than populations of the nine-spined stickleback. First, we found higher *P*_ST_ values for 13 out of 15 traits in three-spined stickleback, with only the length of the first branchial arch and the length of the first dorsal spine showing stronger divergence in nine-spined stickleback (Figure 2). Second, *P*_ST_ values had a range of 0.037–0.49 (average = 0.24 ± 0.14) and 0.013–0.27 (average = 0.10 ± 0.075) for three-spined stickleback and nine-spined stickleback, respectively.

For both species, the first two relative warps explained more than 50% of the variation in body shape, with successive RW scores only increasing the total variation explained by 12.67% or less. In line with the other morphological traits, the diversification of RW1 and RW2 was larger in three-spined stickleback than in nine-spined stickleback, although the differences in *P*_ST_ values were smaller than for the other morphological traits (Figure 2).

### 3.2. Population Structure and Genetic Diversity Statistics

*Genetic diversity*—The genetic diversity per population, calculated as H_E_, showed similar patterns for both species, with generally lower genetic diversity for populations that are further away from the coast (Table 1; Figure 3). However, both LFW and UFW populations showed similar levels of H_E_ in three-spined stickleback, while UFW populations had substantially lower H_E_ levels than LFW populations in nine-spined stickleback. As a result, the relationship between genetic diversity and distance to the coast was only significant in nine-spined stickleback (3s: slope = −0.00020; *P* = 0.242; 9s: slope = −0.00076, *P* = 0.0002) (Figure 3).

*Genetic differentiation*—Overall neutral *F*_ST_ was 0.102 and 0.086 in three-spined stickleback and nine-spined stickleback, respectively, and a significant isolation-by-distance pattern was observed in both species (3s: r = 0.42, *P* = 0.0487; 9s: r = 0.72, *P* = 0.0055) (Appendix A). Based on the maximal marginal likelihood scores, the most likely number of clusters to explain the population structure was K = 6 for both species (Figure 4 and Figure 5). Yet, in three-spined stickleback there was strong support for population structuring between populations from LBW and LFW sites (Figure 4). Nine-spined stickleback showed less population divergence among populations from the LBW and LFW sites, while populations from UFW sites were genetically more isolated (Figure 5). MDS plots (Appendix A) based on pairwise *F*_ST_ (Appendix A) confirmed the Bayesian structure analyses.

### 3.3. Signatures of Adaptive Divergence

*Morphological signatures of selection—*On all four geographical scales, the proportion of morphological traits for which *P*_ST_ significantly exceeded *F*_ST_ was higher in three-spined stickleback (23–41%) than in nine-spined stickleback (6–12%; Table 2). The morphological traits that showed evidence for adaptive divergence largely overlap among the four geographical scales and the two species (Appendix A). In three-spined stickleback, the lowest number of significant *P*_ST_ values were found on the LFW–UFW scale (4/17 traits). On the other geographical scales, three-spined stickleback showed equally strong (7/17) signals of adaptive divergence, primarily due to stronger divergence in the length of gill rakers (traits LGR2, LGR3 and LGR4). In nine-spined stickleback, signals of adaptive divergence varied from 1/17 to 2/17 traits across the four geographical scales considered (Table 2).

*Genomic signatures of selection—*Our first outlier detection approach using BayeScan identified 142 and 70 outliers across the eleven sites in three-spined and nine-spined stickleback, respectively. (Figure 6). Across all geographical scales, the proportion of outlier SNPs was higher in three-spined stickleback (0.77–1.31%) than in nine-spined stickleback (0.21–0.46%) (Table 3). In three-spined stickleback, the proportion of outliers was comparable across the sides of the LBW–LFW–UFW triangle (Table 3). In contrast, in nine-spined stickleback, the proportion of outlier SNPs was markedly higher across the LBW–UFW side of the triangle (Table 3). Our second outlier detection approach using GRoSS v1.0 was more stringent and identified an overall lower proportion of outlier SNPs. Yet, the strongest signal of adaptive divergence was again found in three-spined stickleback (3s: 0.0383%; 9s: 0.0333%). The overall genomic signatures of selection showed largely congruent patterns for the two species with respect to the distribution of outlier SNPs along the branches of the admixture graph. First, the majority of outlier SNPs were detected along the local site-specific branches (3s: 71/83; 9s: 92/100) of the admixture graph. Second, the majority (3s: 67/71; 9s: 80/92) of these outlier SNPs were detected along branches specifically associated with the freshwater sites, including both the LFW and UFW sites (Figure 7). Finally, only few outlier SNPs (N = 2 in both species) were detected along the branch splitting the freshwater from the brackish water populations (w-s). These outlier SNPs were associated with linkage group IV in three-spined stickleback (Appendix A), and with linkage groups V and XII in nine-spined stickleback (Appendix A). These genomic positions were also detected as outliers in the BayeScan v2.1 analyses carried out on the geographical scales that included a brackish water—freshwater transition. However, GRoSS v1.0 also revealed some dissimilarities in genomic signatures of selection between the two species. In the nine-spined stickleback, the LBW sites accounted for a larger proportion of outlier SNPs than in three-spined stickleback. Additionally, the relative contribution of outlier SNPs associated with lowland and upland freshwater sites was found to be reversed. In three-spined stickleback the largest proportion of outlier SNPs was associated with LFW sites, while in nine-spined stickleback the largest proportion of outlier SNPs was associated with UFW sites (Figure 7).

## 4. Discussion

In this study, we investigated the population structure, levels of divergence and signatures of selection in coexisting three-spined and nine-spined stickleback populations from Belgium and the Netherlands. In each species, we compared three geographic contexts, i.e., a short-range brackish water-freshwater transition (LBW–LFW), a long-range brackish water-freshwater transition (LBW–UFW), and long-range spatial isolation (LFW–UFW). Since selection is expected to be strong across the brackish water-freshwater transitions (LBW–LFW and LBW–UFW), and gene flow is expected to be weak across the long-range comparisons (LBW–UFW and LFW–UFW), our sampling design allowed us to assess the single and joint effects of selection and gene flow on genetic and morphological variation in both species. While the two species share evident similarities in terms of population genetic diversity and structure, we found clear differences in the observed levels of adaptive divergence across the three geographical contexts and the two species. First, morphological and genomic adaptive divergence in three-spined stickleback was strong, in any geographical context. Second, morphological and genomic adaptive divergence in nine-spined stickleback was weaker, but the strongest genomic signatures of selection were observed across the long-range brackish water—freshwater transition (LBW–UFW). Third, for both species signals of positive genomic selection were low or missing for the genealogical branches associated with brackish water sites (LBW), while these signals were stronger for the genealogical branches associated with freshwater sites (LFW and UFW).

The locations in this study vary in salinity, which is known to impose important selection pressures in both the three-spined stickleback [21,33,38] and the nine-spined stickleback [53,54]. Thus, we expected salinity to be one of the important common drivers of adaptive divergence among the populations of both species. Accordingly, we generally observed stronger signatures of adaptive divergence in both species when contrasting populations from brackish water sites (LBW) with populations from freshwater sites (LFW and UFW), than among populations that were isolated by distance, but not by a different salinity environment (LFW vs. UFW). Most outlier SNPs in the two species were assigned to local terminal branches, and most of those involved freshwater sites. This may indicate that freshwater habitats are ecologically heterogeneous, imposing diverse selection pressures on local populations of both species. For both species, only two outlier SNPs were assigned to the root of the split between brackish water and freshwater populations (w-s). These outliers were located on different linkage groups (LG) for the two species (3s: LG IV; 9s: LG V and LG XII). In three-spined stickleback, one of these outlier SNPs is located within a gene (ID: ENSGACG00000018958) previously described as a candidate for adaptation to variation in salinity [38], while this does not apply to the gene associated with one of the two outlier SNPs in nine-spined stickleback. It has been demonstrated previously that freshwater adaptation in three-spined and nine-spined stickleback has different genomic origins [53,55]. However, the comparable distribution of outlier SNPs along the branches of the population admixture graph, may indicate that selection can leave similar genomic signatures on species that are subject to the same environmental contrast and to similar degrees of spatial isolation.

In line with our previous assessment of adaptive divergence in the two species [22], the most obvious difference between the two stickleback species was the overall stronger signals of adaptive divergence in the three-spined stickleback. We here confirm this observation at three different geographical scales. Thus, the stronger tendency of three-spined stickleback to adapt to local selective environments seems to be independent of the geographical and environmental context. However, our comparison of the three sides of the triangle revealed another important difference between the two species. In the three-spined stickleback, a comparable proportion of outlier SNPs on the LBW–UFW and the LBW–LFW scale indicates that enhanced gene flow at the LBW–LFW scale does not disrupt the effects of selection, and might even promote local adaptation [3]. In contrast, in the nine-spined stickleback, the proportion of outlier SNPs at the LBW–UFW scale was considerably higher than on the LBW–LFW scale, suggesting that gene flow among the LBW–LFW scale may constrain adaptive divergence [3]. In summary, the two species seem to differ in evolutionary potential in the face of high gene flow. They are therefore positioned differently at the migration-selection balance [56], with three-spined stickleback more tilted towards selection, and nine-spined stickleback towards migration [57].

Patterns of phenotypic and genetic divergence might be influenced by past demographic processes [58], thereby limiting our ability to compare the relative importance of recent selection, gene flow and genetic drift in both species. However, our analyses suggest that historical patterns are unlikely to be the dominant driver of the observed levels of phenotypic and genetic divergence in our study system. First, the brackish water and freshwater sites in this system are likely of a relatively young postglacial origin [59], and phylogeographic studies in Europe suggest that both species colonised these areas following the retreat of the Late Pleistocene ice sheets [60,61]. Such postglacial origins also have been confirmed for three-spined stickleback in our study area [62]. Second, the two species show clear similarities in population structure, with evolutionary relationships among populations that correspond well with the three defined spatial scales (LBW, LFW, UFW). More recently, aquatic habitats in Belgium and the Netherlands have been influenced by strong anthropogenic change in the form of the construction of canals, dykes and drainages. This has likely affected the distribution and genetic diversity of both species in comparable ways.

Importantly, a similar population structure in the two species implies that we may expect similar performance of outlier detection methods, both in detecting genomic signatures of selection based on departures from a baseline model under genetic drift, as well as in avoiding false positives. The various outlier detection methods used in this study (BayeScan, GROSS) as well as in our previous assessment of adaptive divergence in the two stickleback species (LOSITAN, Arlequin, BayeScan) [22] have consistently revealed a higher proportion of outliers in three-spined stickleback than in nine-spined stickleback, in line with the stronger morphological divergence in this species. Nevertheless, a recent view is that without estimates of local recombination rate, interpreting genome scan results is difficult [63]. Caution with the interpretation of the results is thus warranted until the incorporation of estimates of local recombination rate become feasible for non-model species.

The difference in evolutionary potential between both species is of particular significance because we compare them in exactly the same environmental matrix. Yet, it is important to remind that the absence of a strong signature of adaptive divergence in the nine-spined stickleback does not imply that the populations of this species are not adapted, since they might already be preadapted to the ecological gradients in the landscape [22]. Additionally, it is important to acknowledge that nine-spined stickleback may harbour stronger signals of adaptive divergence for other forms of variation that were not assessed in this study. These forms of variation might, for example, be expressed by differential epigenetic patterns or copy number variations in genomic regions under selection. However, just focusing on our current findings, there are various species-specific properties such as genomic architecture, dispersal capacities and life history that may underlie the different evolutionary responses in the two species that have been discussed previously [22]. 

However, in this study, we could also shed light on how these species-specific properties may lead to disparate evolutionary dynamics at different geographical scales. In particular, three-spined stickleback, genetic diversity was highest for the LBW populations, and comparably lower for the LFW and UFW populations. In contrast, genetic diversity in nine-spined stickleback was similar for LBW and LFW populations, but substantially lower for the UFW populations. This pattern suggests that the two species are unequally affected by genetic drift in different parts of the landscape. In LFW sites, nine-spined stickleback populations are possibly more resilient than the three-spined stickleback populations, as they may be able to cope better with hypoxic conditions during summer droughts [22,64]. In UFW sites, however, we speculate that three-spined stickleback populations are more stable than nine-spined stickleback populations, because their better swimming capacity might enable them to better cope with high flow rates following heavy rainfall [24,65]. In summary, species-specific properties may interact with regional environmental factors that differentially affect population persistence in the two species.

## 5. Conclusions

In conclusion, our landscape-level comparison between two closely related coexisting species revealed that the interaction between species-specific properties and regional landscape features might lead to distinct evolutionary responses and levels of adaptive divergence. This finding has profound implications for conservation biology and biodiversity management, as it emphasises that although adaptive potential is to a large extent species-specific, how much adaptive potential is realised may vary throughout the landscape.

## Figures and Tables

**Figure 1 genes-12-00435-f001:**
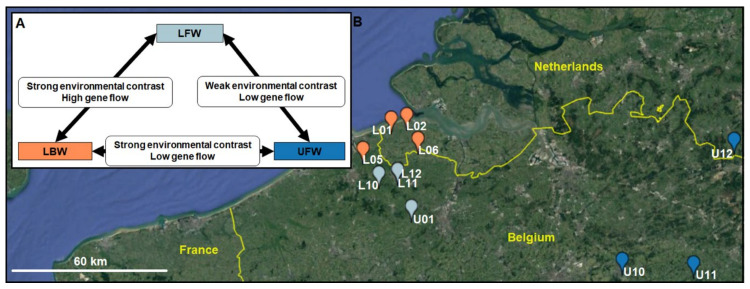
(**A**) Triangle representing the contrast among the three geographical scales. (**B**) Overview of the study area, lowland brackish water (LBW), lowland freshwater (LFW), and upland freshwater (UFW) are represented by orange-red, light-blue and dark-blue dots, respectively.

**Figure 2 genes-12-00435-f002:**
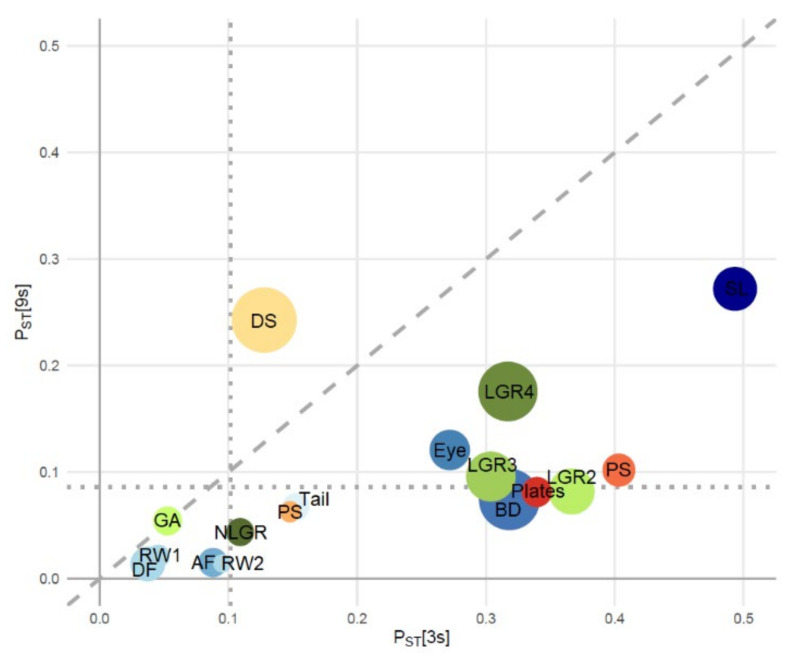
Levels of phenotypic differentiation (*P*_ST_) among three-spined ([3s]) and nine-spined ([9s]) stickleback populations for 17 morphological traits (see Section 2.2 for description of trait codes), with the diagonal dashed line representing the 1:1-line. The dotted vertical and horizontal lines represent the level of neutral genetic divergence in [3s] (*F*_ST_ = 0.102) and [9s] (*F*_ST_ = 0.086), respectively. Circle sizes indicate the importance of parallel versus non-parallel effects. These effect sizes were determined using ANOVAs attributing the variation in each trait to the effect of site (degree of parallelism), the effect of species, and the effect of site by species interaction (degree of non-parallelism). Shades of *red-orange*, *blue* and *green* represent traits related to body armour, body shape and gill morphology, respectively.

**Figure 3 genes-12-00435-f003:**
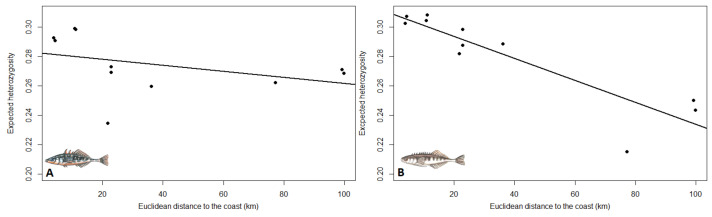
The relationship between Euclidean distance to the coast and expected heterozygosity in three-spined stickleback (A; slope = −0.00020, *P* = 0.242) and nine-spined stickleback (B; slope = −0.00075, *P* = 0.0002).

**Figure 4 genes-12-00435-f004:**
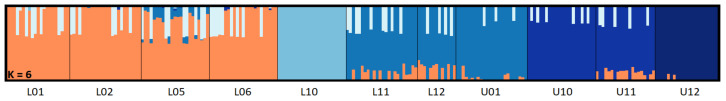
Barplots of population structure with K = 6 clusters in three-spined stickleback inferred by Bayesian analysis using fastStructure v1.0.

**Figure 5 genes-12-00435-f005:**
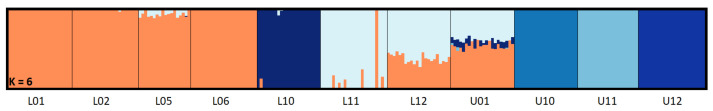
Barplots of population structure with K = 6 clusters in nine-spined stickleback inferred by Bayesian analysis using fastStructure v1.0.

**Figure 6 genes-12-00435-f006:**
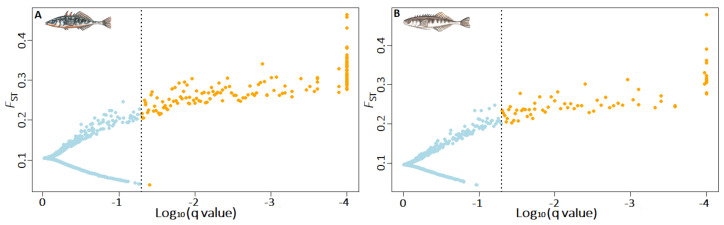
Proportion of outlier SNPs across the geographical scale LBW-LFW-UFW (eleven sites) detected by BayeScan v2.1 for (**A**) 10,836 SNPs in three-spined stickleback and (**B**) 15,033 SNPs in nine-spined stickleback.

**Figure 7 genes-12-00435-f007:**
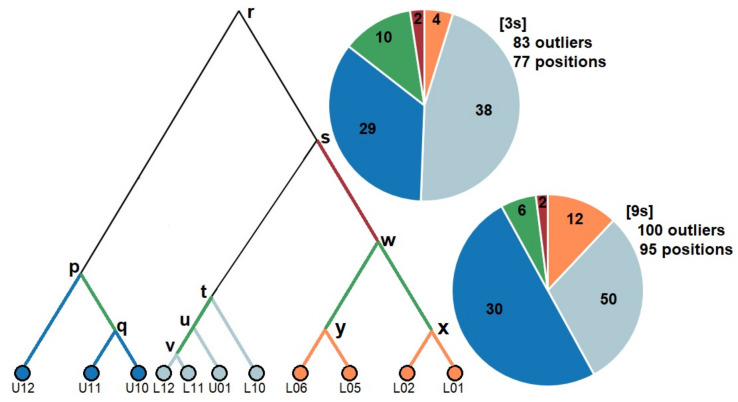
Consensus admixture graph used by GRoSS v1.0 for the detection of outlier SNPs in the two species, along with pie charts visualising the distribution of outlier SNPs across five categories of specific branches of this admixture graph. *Orange*: lowland brackish water terminal branches. *Light blue*: lowland freshwater terminal branches. *Dark blue*: upland freshwater terminal branches. *Green*: branches within the geographical subgroups. *Dark red*: branch splitting the freshwater from the brackish water sites. In total, the 10,836 (3s) and 15,033 SNPs (9s) were tested for signatures of selection along 20 branches, resulting in a proportion of 0.0383% (83/[20*10,836]) and 0.0333% (100/[20*15,033]) outliers in three-spined stickleback and nine-spined stickleback, respectively. The distribution of outlier SNPs over the five categories differs significantly between species (Chi-squared test: χ^2^ = 10.03, *P* = 0.040).

**Table 1 genes-12-00435-t001:** Overview of population genomics statistics per site for populations of three-spined ([3s]) and nine-spined ([9s]) stickleback. DTC: distance to the coast in kilometers, *n*: number of individuals per population retained for downstream analyses, *Ho*: observed heterozygosity, *He*: expected heterozygosity, *F_IS_*: inbreeding coefficient. *N_e_*: effective population size.

Site	DTC	*n*[3s]	*Ho*[3s]	*He*[3s]	*F_IS_*[3s]	*N_e_*[3s]	*n*[9s]	*Ho*[9s]	*He*[9s]	*F_IS_* [9s]	*N_e_*[9s]
**L01**	3.94	21	0.266	0.292	0.0805	213.4	22	0.246	0.303	0.167	441.7
**L02**	4.30	24	0.266	0.290	0.0769	166.4	23	0.299	0.307	0.032	429.7
**L05**	10.90	23	0.282	0.299	0.0550	198.4	18	0.240	0.304	0.187	142.0
**L06**	11.14	23	0.294	0.298	0.0186	166.5	23	0.299	0.309	0.031	185.0
**L10**	21.75	23	0.229	0.234	0.0249	120.8	22	0.266	0.282	0.052	690.2
**L11**	22.84	24	0.229	0.273	0.141	72.2	23	0.291	0.288	0.001	127.4
**L12**	22.84	13	0.159	0.269	0.344	814.0	22	0.283	0.299	0.052	508.3
**U01**	36.20	24	0.232	0.259	0.097	219.5	22	0.283	0.289	0.022	575.5
**U10**	77.10	23	0.239	0.262	0.081	137.2	22	0.163	0.215	0.213	177.1
**U11**	99.20	20	0.261	0.271	0.034	138.8	21	0.219	0.250	0.114	344.5
**U12**	99.80	21	0.253	0.268	0.053	94.8	23	0.175	0.244	0.251	271.2

**Table 2 genes-12-00435-t002:** Proportion of *P*_ST_ values that significantly exceed neutral genetic divergence *F*_ST_ at four geographical scales. Included are the 14 morphological traits (excluding standard length) and relative warp 1 and relative warp 2 defining variation in body shape. LFW: lowland freshwater, LBW: lowland brackish water, and UFW: upland freshwater.

Geographical Scale	*F_ST_* [3s]	Mean *P_ST_* [3s]	*P_ST_* > *F_ST_* [3s]	*F_ST_* [9s]	Mean *P_ST_* [9s]	*P_ST_* > *F_ST_* [9s]
**LBW–LFW–UFW**	0.102	0.22	7/17 (41%)	0.086	0.090	2/17 (12%)
**LBW–LFW**	0.081	0.19	7/17 (41%)	0.040	0.071	1/17 (6%)
**LBW–UFW**	0.071	0.20	7/17 (41%)	0.097	0.093	2/17 (12%)
**LFW–UFW**	0.140	0.19	4/17 (23%)	0.118	0.100	1/17(6%)

**Table 3 genes-12-00435-t003:** Proportion of outlier SNPs on four geographical scales as detected by BayeScan v2.1 for 10,836 SNPs in three-spined stickleback ([3s]) and for 15,033 SNPs in nine-spined stickleback ([9s]). LFW: lowland freshwater, LBW: lowland brackish water, and UFW: upland freshwater.

Geographical Scale	Total Outliers [3s]	Proportion [3s]	Total Outliers [9s]	Proportion [9s]
**LBW–LFW–UFW**	142	1.31%	70	0.46%
**LBW–LFW**	98	0.90%	44	0.29%
**LBW–UFW**	87	0.80%	61	0.41%
**LFW–UFW**	83	0.77%	31	0.21%

## Data Availability

Raw data, VCF files and morphological measurements are available upon request. Data files containing all outlier SNP positions are available online at: https://doi.org/10.6084/m9.figshare.14210651.v1 (uploaded on 12/03/2021).

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
