# Peer review of "Adaptive Divergence under Gene Flow along an Environmental Gradient in Two Coexisting Stickleback Species"

_genes, 2021, doi:10.3390/genes12030435_

Round 1

Reviewer 1 Report

I have now reviewed the manuscript “Adaptive divergence under gene flow along an environmental gradient in two coexisting stickleback species”. The authors proposed to investigate the role of migration, approximated by geographic distance, and selection, approximated by freshwater vs brackish water habitats, as drivers of genetic divergence across a spatial scale while utilizing a pair of closely related species as natural replicates of emerging adaptive patterns. The manuscript holds an interesting and clean story, and the clarity of the writing greatly facilitates its understanding. The background, research questions and hypothesis are also described in a very concise manner. I fully agree with the interpretation of the results and subsequent discussion.

Still, I feel that a more decisive assessment of the interplay between selection (by habitat-type) and migration (by geneflow estimates) in shaping the observed patterns of adaptative divergence is missing.  This is because not only outlier detection methods are sensitive to population structure but also that a) it is reported a strong pattern of IBD but also b) the number of detected clusters (K=6) is higher than the geographical scale at which outlier tests were performed, which only considered 4. Sub-structuring could be affecting the outputs of outlier detection strategies, especially if the magnitude of population differentiation, i.e. outlier´s allelic frequency among populations, would not equally distributed among cluster pairs. I wonder if a redundancy analysis utilizing only the outliers (allelic frequencies per location) versus environment (salinity) and geographic distances (latitude, longitude, for instances) could help to resolve this potential issue. 

Other minor issues:

Lines 58 -59. Please clarify what “specific thresholds” you refer to.

Line 181: Considering the role of copy number variation on three-spine stickleback evolution, could this unexplored line of research (by removing potential paralogs) hinder a generalization of the findings reported for these species? For instances, do the outliers detected in the study represent the full extent of adaptive divergence, or expanding the divergence “screening” to CNV would lead to different patterns, i.e. higher divergence among nine-spines?  I feel this possibility should be added to the discussion.

Line 224: Prior odds of 100 in Bayescan are bit too low, with the program manual suggesting 10- and 100- fold orders of magnitude higher values to obtain robust outliers. I wonder if at least some experimentation with pr=1000 or pr=10000 were attempted a priori before choosing 100, and then, what is the rationale to maintain such a low pr.

Lines 224-on: In the methods says that Bayescan was performed globally and for each site, while for GROSS the dataset was further divided into species. However, results report Bayescan outliers for both three- and nine-spine sticklebacks. Please clarify in the methods.

Line 250: Please remove “clearly”

Line 276: Please define “m”

Lines 415-418: I think this paragraph break is incorrect. I would rather break at line 415.

Fig. 2 – This figure is confusing. For example, the code for the circles´ colour is not given nor explained in the caption and the default grey background somehow overshadows the dotted lines – almost look like a third axis on the graph.

Fig. 7 – This figure also deserves a better editing. Once again, the default grey background does not help to contrast the colour choice for three-spines. The font size should also be enlarged. Lastly, what do the LXX_ and UXX_ on the x-axis of B represent? – should be added to the caption.

Reviewer 2 Report

The study untitled “Adaptive divergence under gene flow along an environmental gradient in two coexisting stickleback species” describes the population genetic structure and morphological differentiation of sympatric tree-spined and nine-spined stickleback and aims to investigate the spatial and environmental drivers of population divergence. This study is very similar to a previous one (Raeymaekers et al. 2017, Nat. Commun.), except the addition of a third set of more isolated sites. Thus, this new study design enables to assess the effect of gene flow and environmental selection on genetic and morphological divergence.

Overall, I find the paper interesting, the methods used and analyses performed seems to be well conducted and the results are clearly described. 

My main concerns are about the need of clarification at some methodological points.

Line 165: Can the authors provide a rational in the use of methylation sensitive restriction enzyme and explain how this measurement of genetic variation is not biased given the environmental influence on DNA methylation variation? Response to different environments (e.g. freshwater vs brackish water, as included within this study) and genetic background can result in DNA methylation variation among individuals. However, a methylated locus will not be cut by a methylation-sensitive enzyme, and thus, no sequencing read would be obtained for this specific locus. This could therefore result in a bias in the estimation of allele frequency and genetic variation.

Figure 2 legend: To what parallel vs non-parallel effects refer to?

Line 358: Add “the single and joint effects of selection and gene flow [on genetic and morphological variation]”.

Line 424: missing a “)”

Line 436: Reference format

Finally, the interpretation made at lines 371-375 (“Thus, we expected salinity to be one of the important common drivers of adaptive divergence among both species.”) is not well supported, given the results presented in tables 2 and 3: is 87/10836 SNPs significantly higher than 83/10836? Similarly, in lines 323-324, it is stated that “the strongest signal of adaptive divergence was again found in three-spined stickleback (3s: 0.037 %; 9s: 0.031%; Figure S4 and S5).”, while the significate difference between 0.037% vs 0.031% is not convincing again.

I’m not saying that the currently used comparison is wrong, but I would suggest another types of statistical analyses to have a more quantitative and straightforward comparison. The study design is well thought to assess the single and joint effects of salinity and gene flow on both morphological and genetic variation, but the authors seems to no take advantage of that in their statistical design. For example, I would suggest a variance partitioning (e.g. redundancy analysis) to clearly quantify the effect of gene flow, salinity, interaction and even the confounding effect of both gene flow and salinity on the measured genetic (neutral or outlier loci) and morphological variation.
